# Endothelial Senescence: From Macro- to Micro-Vasculature and Its Implications on Cardiovascular Health

**DOI:** 10.3390/ijms25041978

**Published:** 2024-02-06

**Authors:** Peichun Wang, Daniels Konja, Sandeep Singh, Beijia Zhang, Yu Wang

**Affiliations:** 1State Key Laboratory of Pharmaceutical Biotechnology, The University of Hong Kong, Hong Kong SAR, China; peichunw@connect.hku.hk (P.W.); kdaniels-1@outlook.com (D.K.); sandeep886397@gmail.com (S.S.); u3591923@connect.hku.hk (B.Z.); 2Department of Pharmacology and Pharmacy, LKS Faculty of Medicine, The University of Hong Kong, Hong Kong SAR, China

**Keywords:** endothelial cells, senescence, microcirculation, senolytics

## Abstract

Endothelial cells line at the most inner layer of blood vessels. They act to control hemostasis, arterial tone/reactivity, wound healing, tissue oxygen, and nutrient supply. With age, endothelial cells become senescent, characterized by reduced regeneration capacity, inflammation, and abnormal secretory profile. Endothelial senescence represents one of the earliest features of arterial ageing and contributes to many age-related diseases. Compared to those in arteries and veins, endothelial cells of the microcirculation exhibit a greater extent of heterogeneity. Microcirculatory endothelial senescence leads to a declined capillary density, reduced angiogenic potentials, decreased blood flow, impaired barrier properties, and hypoperfusion in a tissue or organ-dependent manner. The heterogeneous phenotypes of microvascular endothelial cells in a particular vascular bed and across different tissues remain largely unknown. Accordingly, the mechanisms underlying macro- and micro-vascular endothelial senescence vary in different pathophysiological conditions, thus offering specific target(s) for therapeutic development of senolytic drugs.

## 1. Introduction

Cellular senescence is the cessation of cell division. Senescence is originally defined as the irreversible loss of proliferative potential in somatic cells, which enter a viable and metabolically active state of permanent growth arrest that is distinct from quiescence and terminal differentiation [1,2,3]. It is characterized by a number of distinct phenotypic changes, such as enlarged cell size, elevated senescence-associated beta-galactosidase (SA-β-gal) activity, formation of telomere-associated foci, and increased expression of cyclin-dependent kinase inhibitors p21 and p16 [4,5,6,7,8]. SA-β-gal activity is commonly used as a biomarker for cellular senescence. Lysosomal beta-galactosidase activity is ordinarily detected at low pH (about pH 4) but becomes detectable at a higher pH level (pH 6) in senescent cells due to significant expansion of the lysosomal compartment [9]. Cellular senescence occurs in not only mitotic, but also postmitotic cells, referred to as the replicative and premature types, respectively [10,11]. Premature senescence occurs upon exposure to various stress conditions, including DNA damage [12], reactive oxygen species (ROS) [13], oncogene activation [12,14], telomere attrition [15,16,17], and metabolic dysregulation [18]. Accumulation of senescent cells contributes to age-related tissue degeneration by developing a complex senescence-associated secretory phenotype (SASP) [19,20,21,22,23,24]. By secreting a plethora of factors, including pro-inflammatory cytokines, chemokines, growth modulators, matrix metalloproteinases, and compromised extracellular vesicles which represent senescence-associated phenotype, senescent cells reprogram the surrounding microenvironment and cause tissue damage, thus promoting ageing and the development of age-associated diseases [20,24,25,26,27,28,29]. Intervention experiments have proven that senescent cell accumulation is an important driver of age-associated functional decline, (multi-)morbidity, and mortality [8]. Systemic clearance of senescent cells delays ageing and extend lifespan [8]. Therapeutically targeting cellular senescence, known as senotherapy, to eliminate senescent cells or induce senolysis, represents a rapidly growing and promising strategy for the prevention and/or treatment of ageing-related diseases [8,30,31,32,33]. Targeting senescent cells can improve both health- and life-span in mice [34]. In both pre-clinical and clinical models of geriatric decline and chronic illnesses, the efficacy of senescent cell removal via apoptosis-inducing “senolytic” drugs or therapies that inhibit the senescence-associated secretory phenotype, SASP inhibitors, has been demonstrated. The present review focuses on the senescence of endothelial cells at different anatomic locations of the vasculature and the implications in macro- and micro-circulatory diseases. The prospect and challenges of anti-endothelial senolytic therapy will also be discussed.

## 2. Microvascular Circulation and CVD

The microvasculature is a broad term used to describe blood vessels with diameters of 100 µm or less. In the peripheral circulation, the cardiovascular system terminates into a network of micro-vessels such as arterioles, capillaries, venules, and other cellular components all of which function to meet the oxygen and nutrient requirements of cells and tissues [35,36]. In addition, these vessels play an important role in peripheral vascular resistance and blood pressure regulation, as well as contributing significantly to immune function by trafficking lymphocytes and leukocytes to target tissues. Thus, these vessels are usually in direct contact with cells of body tissues, making the microvasculature arguably the most important aspect of the cardiovascular system. Vessels that make up the microcirculation are almost entirely lined with endothelial cells which together with smooth muscle layer play an important role in regulating the tone of arterioles [36] via three mechanisms: myogenecity, metabolic control, and neuro-hormonal regulation [37]. Under physiological conditions, these three mechanisms ensure progression and maintenance of proper microcirculatory function. However, under conditions of disease, dysregulation of these control mechanisms results in microvascular dysfunction.

### 2.1. Ageing and Microvascular Dysfunction

Biological age is reflective of vascular age, and microvascular age is cardinal in the ageing process as it exerts immense influence on every tissue/organ of the body [38]. Thus, a decline in microvascular function is a hallmark of the biological ageing process [39]. Endothelial dysfunction is an indicator of microvascular ageing, and age-related endothelial dysfunction refers to an impairment in vasodilatation or reduction in the production of endothelium-derived nitric oxide, as a result of endothelial cell senescence [40]. As a result of decreased endothelial NO synthase expression and reduced NO production, inducible NO synthase (iNOS) and ROS levels increase significantly, leading to DNA damage and promotion of senescence and apoptosis [39,41]. In addition, aged vascular endothelium have reduced ability to release growth factors [42], reduced regenerative capacity [43], and impaired tissue angiogenesis [44]. Low-grade inflammation is characteristic of aged microvasculature which affects both the endothelial layer as well as the surrounding layers such as the perivascular adipose tissue. The low-grade inflammation further disrupts the homeostatic capacity of the microvasculature and promotes microvascular disease [38]. Decline in mitochondrial function, dynamics, and metabolic activity due to ageing is significantly implicated in many ageing-associated diseases. Mitochondrial dysfunction, classically referring to impaired electron transport and decreased ATP production, typically results in increased mitochondrial ROS (mROS) production [38]. Sustained increased mROS production leads to mitochondrial DNA damage and accelerated ageing. In the vascular bed, endothelial senescence and smooth muscle dysfunction are subsequent to mitochondrial dysfunction and increased mROS. Increasing evidence of the role of mitochondrial dysfunction and oxidative stress in accelerated ageing has opened new paradigms and offered insights into potential therapies for microvascular dysfunction. Thus, targeting oxidative stress in the microvascular bed may represent a promising therapeutic strategy to slow down endothelial senescence and accelerated vascular ageing.

### 2.2. Microvascular Dysfunction in Obesity

Evidence of microvascular dysfunction in obesity is abound, and these largely point to endothelial dysfunction in the microvasculature of obese subjects. For example, a human study using skin and resistance arteries obtained from obese patients demonstrated that arteries responded poorly to endothelium-dependent vasodilatory agents, including the all-important insulin-induced endothelium-dependent vasodilatation [37]. In addition, in animal models, rarefaction of capillary density and structural remodeling of micro-vessels have been demonstrated [37]. Thus, obesity alters microvascular structure and function, leading to breakdown on the regulation of vascular tone and blood pressure regulation. In the heart, for instance, obesity is detrimental to coronary microcirculation. Development of coronary artery disease as a result of structural and functional changes in the coronary vasculature is often seen among obese patients [45]. Studies have shown that myocardial ischemia often results from coronary microvascular abnormalities in obesity. Although there is controversy surrounding the reduced myocardial perfusion in obesity, there is general consensus of reduction in response to vasodilator agents by coronary resistance arteries. This may either contribute to cardiovascular disease risk or to obesity pathogenesis [45]. The mechanism by which obesity causes reduction in vasodilator activity in coronary micro-vessels, however, remains to be properly elucidated. In addition, increased adiposity around the heart and coronary arteries shifts the balance of adipokine secretion. Many studies have shown that adipokines such as adiponectin, leptin, and resistin have direct effects on the vasodilatory function of coronary arteries and arterioles on obesity. Whereas some of these studies have pointed to endothelial dysfunction and subsequent loss of nitric oxide production as a mediator of adipokine-induced coronary artery dysfunction in obesity, other studies point to increased ROS production in the adipocytes and coronary vasculature [46,47,48,49]. However, the exact mechanism by which adipokines induce increases ROS production still needs to be studied.

### 2.3. Microvascular Dysfunction in Diabetes Mellitus

Arteriosclerotic lesions are cardinal events in type-2 diabetes mellitus (T2DM) patients. In addition to this, risks of myocardial infarction and cerebrovascular accidents are more likely in T2DM patients compared to in non-diabetic patients [50]. Cardiac and vascular complications of diabetes, including micro- and macro-angiopathy, are the leading causes of death among T2DM patients [51,52]. Dysfunction in the coronary microvasculature and impaired vasodilatation are characteristic of T2DM. Humans with T2DM generally have impaired microvascular endothelial function, reduced NO, and enhanced ROS production [53,54]. There is a consensus that increased mitochondrial ROS production in endothelial cells of T2DM patients promotes microvascular damage [55]. Endothelial cells typically respond to hyperglycemia by increasing ROS production. Subsequently, ROS-induced DNA damage activates DNA repair enzyme poly[ADP ribose]polymerase 1, which initiates a cascade of events, eventually leading to onset and/or progression of vascular injury [56]. Importantly, while ECs in culture respond to hyperglycemia-induced ROS stress within short periods and animal models show microvascular injury within weeks of hyperglycemia, human T2DM subjects require many years to develop microvascular damage. The explanations of the long duration of hyperglycemic exposure leading to anatomical damage to the microvasculature is currently a subject of intense investigation [56].

### 2.4. Microvascular Dysfunction in COVID-19

CVD risk factors (age, sex, obesity, diabetes mellitus, and hypertension) are also documented risk factors for COVID-19 [57]. Microvascular endothelial cell damage is reportedly a hallmark of COVID-19 progression subsequent to pro-inflammatory cytokine release after host invasion and immune activation [58,59]. Concomitant damage to the microvascular endothelium leads to sepsis, a severe form endothelial dysfunction syndrome, leading to septic shock and death due to irreversible damage to the microcirculation [60]. COVID-19 patients typically have reduced ACE2 levels which result in reduced vasodilatory activity in the microvasculature. Decreased coronary flow reserve and increased microvascular resistance as a result of systemic inflammation in COVID-19 patients result in coronary microthrombi as a result of injuries to the coronary microvasculature [61,62].

## 3. Endothelial Senescence

The endothelium forms an integral layer at the luminal surface of blood vessel and participates in physiologic functions including wound healing, hemostasis, substance transfer, and vascular tone/reactivity regulation [63,64]. The turnover of endothelial cells has been generally considered very slow. However, in areas of bifurcations and branching points of the arteries, the haemodynamic forces of shear and stretch cause chronic injuries to the endothelium; thus, the rate of endothelial cell replication/regeneration is increased [65]. Endothelial regeneration is of pivotal importance for maintaining the integrity of endothelium, in turn promoting vascular repair and health [66]. Regenerating endothelial cells arise from sites adjacent to the injury and exhibit significant proliferative potential [67]. However, the regenerative response is significantly impaired with age in older subjects due to the development of endothelial senescence, which influences cell responses [68,69]. Endothelial progenitor cells (EPCs) mediate repair mechanisms for endothelial regeneration and maintenance, probably by directly differentiating into endothelial cells (ECs) and integrating into injured vessels, or act on cells and blood vessels by releasing paracrine substances [70]. Aging subjects showed negative effects on the EPC number and function, of which subsequently impair endothelial cell repair and regeneration in the aging vasculature [71,72,73,74], thus indicating that the EPCs are involved in endothelial senescence as well. “Senescence” refers to the finite capacity for division in normal diploid cells. Endothelial cell senescence can be attributed to a number of factors on vascular pathologies, especially oxidative stress and sustained cell replication [75]. From a molecular point of view, the telomere hypothesis explains the mechanism of cellular senescence. Telomeres are the physical ends of chromosomes, and the presence of telomerase, a specialized reverse transcriptase, is required for the synthesis of telomeric DNA. The telomere hypothesis suggests that cells lacking telomerase have shortened DNA, and senescence occurs when telomere strength reaches a critical point [76]. Unlike young endothelium, senescent endothelial cells do not respond to growth factors due to defective signaling pathways. With age, the accumulation of dysfunctional senescent cells and the loss of regenerative capacity together contribute to the loss of arterial function and integrity. Depending on different locations and pathophysiological conditions, senescent endothelial cells exhibit upregulation of p53/p21 or p16/retinoblastoma protein (Rb) pathways, markers associated with activation of the DNA damage response (DDR) (p38 mitogen-activated protein kinase (p38MAPK) and phosphorylated histone 2AX (γH2AX)) (Figure 1), senescence-associated heterochromatin foci, SASP, or increased SA-β-Gal activity [77,78,79].

Much of our current knowledge about the properties of senescent endothelial cells is based on experiments in cultured cells. Vascular endothelial cells retain both proliferative potential and a specialized phenotype in a culture dish [80,81,82]. Human umbilical vein ECs (HUVECs) cultured on collagen gels form confluent monolayers that bind silver at their intercellular border similar to cells in vivo [83]. There are intercellular junctional structures present, such as adherents and tight junctions. In contrast, HUVECs grown on plastic surfaces do not stain with silver. The silver-staining characteristic of endothelial monolayers is related to their in vitro maturation and senescence [83]. The endothelial-cell growth factor (ECGF) delays the premature senescence of HUVECs [84]. Typical endothelial cells isolated from aorta or inferior venae cavae are small, round to polygonally shaped with the diameter ranged from 50 to 70 microns and arranged uniformly [85]. Cultured endothelial cells have a finite life-span. Population doubling (PD) time varies as a function of the cell seeding density. After a certain number of PD, the endothelial subculture density rapidly decreases [86]. Addition of the fibroblast growth factor (FGF) shortens PD time but increases the replicative lifespan of endothelial cultures [87]. When the inoculated endothelial cells do not double after three weeks of frequent refeeding with fresh medium and serum, they are considered to be senescent [88]. As cells senesce, their average attachment area increases more than three folds, ranging from 100 to 200 microns in diameter (Figure 2). Giant senescent endothelial cells reach a diameter of more than 250 microns, with the mean areas as high as 3660 μm^2^ [85]. The protein content also increases dramatically, starting at about 400–600 pg/106 young to 2000–2400 pg/106 senescent endothelial cells. Senescent endothelial cells are usually multinucleated and show no incorporation of [3H] thymidine. The loss of proliferative capacity is accompanied by increased chromosomal aberrations, such as translocations, increased heterochromatin foci, and polyploidy and telomere damage, in senescence endothelial cells [89,90,91]. High-mobility group A (HMGA) proteins or heterochromatin markers, including HP1 and tri-methylated lysine 9 histone H3 (H3K9me3), are recognized as molecular markers of senescence-associated heterochromatin foci and are considered to indicate cellular senescence [92]. In large arteries, as chronological age advances, areas with high endothelial cell turnover are covered by clusters of senescent cells [65]. Senescent cells accumulate in the vessels of patients with atherosclerosis, hypertension, aneurysms, diabetes, and intimal hyperplasia [93]. Endothelial cells that have undergone replicative senescence show an increased size, polymorphic nuclei, flattening, and vacuolization. Compared to their normal counterparts, senescent endothelial cells show increased adhesion to the basement membrane and reduced capacity to align with the direction of blood flow and is causatively implicated in age-related inhibition of endothelium-dependent vasodilatation [94,95]. The presence of multinucleated endothelial cells in aortae in vivo has been confirmed by both scanning and transmission electron microscopy [96]. These senescent endothelial cells exist as colonies in the aortae from elderly subjects with intimal-thickened or advanced atherosclerotic lesions and contribute to further development of atherosclerotic lesions [93,97]. Senescence of the endothelium leads to pathological conditions such as impaired vasodilation, increased vascular stiffness, increased vascular permeability, micro-thrombosis, atherosclerosis, in-stent restenosis, hypertension, ischemic or hemorrhagic disorders, and inflammation [95,98,99,100,101]. However, age-dependent vascular dysfunction caused by senescent endothelial cells is location-specific and mechanism-dependent. The differential rates of vascular aging and diversified organ dysfunction caused by endothelial senescence in different locations will be discussed, particularly focusing on the microvasculature of the brain and the heart.

## 4. Microvascular Endothelial Senescence

Endothelial cells differentiate from a common precursor, angioblast, and present many common morphological and functional features [103]. During different stages of differentiation, they express the angiogenic growth factor receptor VEGFR2 (flk1/KDR), one of the receptors for VEGF, and PECAM-1, followed by Tie-2, Tie-1, and vascular endothelial cadherin. Thus, endothelial cells may exhibit intermediate phenotype(s) in vivo. Heterogeneity is a characteristic feature of endothelial cells [104,105]. For example, Weibel−Palade bodies (WPB) are organelles specifically present in endothelial cells to store the von Willebrand factor (VWF). However, there is a heterogeneous distribution of WPB along the vascular tree: absent from the thoracic aorta, rare in the abdominal aorta, present in myocardial capillaries, and numerous in the inferior vena cava and pulmonary artery [106]. The VWF labeling exhibits the same variation in its distribution along the vascular tree as for its storage organelle. Note that even in the same organ, the endothelia of large and small vessels, veins, and arteries exhibit a complex array of specialized functions and molecular signatures. In brain, the expression of VWF is increased in venous compared to arterial and capillary endothelial cells, whereas the major facilitator superfamily domain containing protein 2a (Mfsd2a) protein is highly abundant in capillary compared to arterial and venous endothelium [107]. The microvasculature is comprosed of arterioles, capillaries, and venules [108,109]. The microcirculation allows the delivery of oxygen and nutrients to meet the energetic demands of local tissues, mainly through regulation of vascular tone, angiogenesis, regulation of hemostasis, inflammation, and vascular permeability [55,110]. The tissue rather than the vessel type contributes to microvascular endothelial cell heterogeneity [111]. Depending on the context and the surrounding microenvironment, microvascular endothelial cells display remarkable heterogeneity and acquire organ-specific properties [112,113]. Endothelial cells from the microvasculature of different organs display specialized properties both in vivo and in vitro. Understanding the heterogeneity in microvascular endothelial senescence is of fundamental importance for developing targeted approaches to prevent age-associated functional decline in different organs.

### 4.1. Brain

The endothelial cells of the brain microvasculature serves as the interface between blood and the central nervous system (CNS), so-called blood−brain barrier (BBB) [114]. The BBB primarily consists of brain microvascular endothelial cells (BMECs) that are sealed together by tight junction proteins. Beyond the vagal afferents and choroid plexus, BMECs are the first CNS cell type exposed to the systemic harmful stimuli. BMECs are embedded in a basement membrane and surrounded by pericytes and astrocytes to restrict transcytosis. Endothelial cells of the BBB exhibit unique biochemical and morphological features such as tight intercellular junctions, few pinocytic vesicles, and no fenestra but abundant mitochondria [115]. The communications of BMECs with other cells of the neurovascular unit, such as pericytes, glial cells, and neurons, play a critical role in neurogenesis and maintenance of normal cognitive function. Age-induced alterations of the transcellular transport machinery and increased senescence in BMECs are potential underlying mechanisms of the endothelial dysfunction that causes the BBB breakdown [116]. With age, the BBB dysfunction is characterized by increased leakiness and impaired transport of molecules such as glucose, amyloid-beta peptide, and xenobiotics. There is an increased ratio of senescent endothelial cells (~10%) in the mouse cerebral microcirculation [117]. These changes at the BBB may be drivers of cognitive dysfunction during aging and may be further impaired in neurodegenerative diseases [118].

When compared to endothelial cells of the peripheral vasculature, tissue-nonspecific alkaline phosphatase (TNAP) activity is highly upregulated in BMECs and shows a continuous and uniform layer of distribution across the plasma membrane, thus being used as a histological marker to detect changes in cerebral microvessel function or morphology [119,120]. TNAP is absent in the endothelial cells of the liver sinusoids. However, skeletal endothelial cells contain a strong TNAP activity, the pattern of which is different from BMECs, showing discontinuous or irregularly scattered distribution across the plasma membrane [120]. BMECs also exhibit a typical perinuclear staining of factor VIII. Heparin is included in cultures to retard growth of factor VIII negative cell types (e.g., pericytes and smooth muscle cells). With passage, markers typically used to identify the BBB, including alkaline phosphatase and γ-glutamyl transpeptidase, declines and increases, respectively. While the permeability of BMECs does not change with passage, receptor-mediated transcytosis is altered [121]. Alterations in the specialized transport systems across the cerebral capillary lead to adverse changes in cerebral and neurotransmitter metabolism.

Cultured BMECs are frequently used to study the function of the BBB [122,123]. They develop confluent, contact-inhibited monolayers with a characteristic polygonal appearance [124]. Primary mouse BMECs become senescent (as defined by SA-β-gal staining, upregulation of p21, DNA damage marked by γH2A.X, and heterochromatin foci) after six complete PD. Senescence can also be induced in vitro through physical or chemical stressors, such as ionizing radiation, chemotherapeutics, and oxidative stress [125]. Upon induction of cellular senescence in culture, BMECs undergo cell cycle arrest and acquire SASP, characterized by increased secretion of pro-inflammatory mediators [126]. Note that senescence is a slow process, taking about three to seven days to develop in response to stressors. Senescent mouse BMECs show increased BBB leakage as evidenced by reduced transendothelial electrical resistance and increased permeability to albumin in comparison to non-senescent/low passage cells [118].

With advanced aging, there is an increased ratio of senescent endothelial cells in the cerebral microcirculation [117,127]. Compared to young animals, the percentage of senescent BMECs in old mouse brain increases from 5.23% to 10.06% as revealed by single-cell RNAseq [117]. The total protein in the brain endothelium has been shown to reduce with increasing age [128]. Senescence of cerebromicrovascular endothelial cells leads to cerebral blood flow dysregulation and the BBB disruption, promoting the pathogenesis of vascular cognitive impairment [117,129,130,131]. Since senescent cells elicit a secretory phenotype, senescence of BMECs within the neurovascular unit contributes to the inflammation associated with neurodegenerative diseases [132]. However, evaluation of senescence-associated beta-galactosidase (SA-β-gal) is not enough to consistently detect senescent endothelial cells within the brain tissue. This is because the BBB, in particular BMECs, primarily exist in a quiescent state, which shares some features of senescent cells, such as increased lysosome content/SA-β-gal activity [133,134]. Other senescence markers are notoriously non-specific to detect using antibodies. Multiple senescence biomarkers within the same cells are usually difficult to detect. As a result, there is an urgent need for novel approaches to identifying, quantifying and characterizing senescent BMECs. Furthermore, it is unknown if aging has specific or global effects on different subtypes of BMECs [111]. A deeper understanding of the heterogeneous nature of BMECs will provide cell type-specific vascular therapies targeting endothelial senescence for neurological disorders.

### 4.2. Heart

The most prevalent cardiovascular disease (CVD) is heart failure (HF), which has a high prevalence among the elderly (https://www-who-int.eproxy.lib.hku.hk/en/news-room/fact-sheets/detail/cardiovascular-diseases-(cvds); accessed on 3 September 2022). The prognosis of HF is unacceptably poor, and there is an urgent need to find better therapies for this condition. In particular, heart failure with preserved ejection fraction (HFpEF), which is characterized by diastolic dysfunction, altered ventricular relaxation and filling, increased stiffness, and concentric cardiac remodeling (fibrosis, inflammation, and hypertrophy), represents one of the most complex and prevalent cardiometabolic diseases in the aging population [135,136,137]. HFpEF leads to pressure, but not volume overload, and is present frequently with comorbidities, such as advanced age, obesity, diabetes, and hypertension [138]. Most currently available drugs for HF show little or no effect on mortality rates in patients with HFpEF, confirmed by numerous clinical trials. Instead, guidelines suggest pharmacological approaches to fluid removal, symptomatic relief (e.g., edema), and management of associated comorbidities (e.g., hypertension and chronic obstructive pulmonary disease (COPD)). Agents used include beta blockers and angiotensin-converting enzyme inhibitors (ACEIs’, angiotensin II receptor blockers (ARBs), mineralocorticoid receptor antagonists (MRAs), etc.) [139].

Vascular endothelial cells comprise ~60% of non-cardiomyocytes in the healthy heart and have a major impact on cardiac heath [140]. Based on the common markers *PECAM1*, *CDH5*, and *VWF*, a large-scale single-cell and single-nucleus RNA-seq study reveals 10 populations including arterial, capillary, capillary-like, venous, endocardial, and lymphatic endothelial cells in the heart [141]. The microvascular capillary represents 57.4% of all endothelial cells in the heart. With age or under pathological conditions, the portions of different cell types in the heart change on an individual basis [140,142,143]. Microvascular dysfunction, rarefaction, and chronic low-grade inflammation have been proposed to play major roles in HFpEF development [144,145,146]. Chronic systemic inflammation, in particular, affects other organs such as kidneys and lungs, leading to sodium retention and pulmonary hypertension, respectively. One of the major mechanisms of systemic inflammation causing HFpEF is that it leads to altered paracrine communication between endothelial cells and surrounding cardiomyocytes [147]. Endothelium-dependent coronary microvascular dysfunction, measured by basal myocardial blood flow or coronary flow reserve (CFR), is present in 29% of HFpEF patients [148]. Microvascular rarefaction increases coronary resistance and contributes to insufficient cardiac perfusion, leading to impairment of myocardial blood flow and oxygen delivery [149]. Patients with HFpEF show elevated inflammatory markers such as interleukin-1 type I receptor (IL-1R), tumor necrosis factor α (TNFα), C-reactive protein (CRP), vascular cell adhesion molecule-1 (VCAM-1), and IL-6 [150]. Coronary microvascular inflammation and stiffening lead to ventricular−vascular uncoupling [145].

Senescent endothelial cells are observed in failing hearts, particularly those with diastolic dysfunction, and contribute to microvascular information and cardiac remodeling [93,151,152,153]. Hutchinson−Gilford progeria syndrome (HGPS) is an accelerated aging syndrome associated with premature vascular disease and death due to heart attack and stroke. Endothelial cells (ECs) differentiated from induced pluripotent stem cells (iPSCs) derived from these patients exhibit senescent hallmarks including replication arrest, increased expression of inflammatory markers, DNA damage, and telomere erosion [154]. The expression of p53 is increased in hearts of patients with heart failure and plays a pathological role in causing cellular senescence [152]. Endothelial p53 signaling suppresses angiogenesis and promotes capillary rarefaction in the failing heart. Activation of p53 in microvascular endothelial cells of the left ventricle induces cardiac inflammation and remodeling [155]. Expression of p53 leads to elevated expression of intercellular adhesion molecule (ICAM)-1 by capillary endothelial cells, which enhances macrophage infiltration and cardiac inflammation. Endothelial p53 depletion improves capillary rarefaction and cardiac function while suppressing cardiac fibrosis and remodeling [156].

Female senescence-accelerated mice on a high-fat and high-salt diet develop HFpEF, characterized by diastolic dysfunction, left ventricular (LV) hypertrophy, left atrial dilatation, and interstitial fibrosis, with reduced exercise capacity and increased lung weight. Endothelial senescence contributes to HFpEF in these animals [157]. Aging seems to cause greater DNA damage and telomere dysfunction in vascular endothelial cells than in smooth muscle cells [158]. Excision repair cross-complementation group 1 (ERCC1), an endonuclease non-catalytic subunit, is an essential component in the DNA damage repairing system [101]. Genetic removal of ERCC1 selectively in endothelial cells causes impaired vasodilatation in coronary arteries and decreased cardiac output, but no microvascular inflammation in the heart [99]. Although genomic instability represents a common feature of aging, the accumulation of mutations varies substantially between tissues. Thus, apart from distinct mechanisms, different microenvironments surrounding senescent endothelial cells may also contribute to variations in the pace of vascular aging among individuals, as well as organs within the same human subject or animal model [159]. Regardless, the available evidence suggests that suppression of endothelial senescence represents a promising therapeutic option for HFpEF, although the inter- and intra-individual heterogeneity should be taken into consideration [160,161].

## 5. Endothelial Senescence and Human Diseases

Endothelial cells are the first group of cells to be directly affected at the onset of senescence due to their location in the vasculature. Senescence affects endothelial cells mainly by decrease in endothelium-associated vasodilators (nitric oxide, hydrogen sulfide, and endothelium-dependent hyperpolarizing factors) and an increase in endothelium-associated contractile factors (e.g., reactive oxygen species, endothelin-1, and angiotensin II). Physiologically, endothelial cells in the vasculature are in a constant process of damage and repair to maintain vascular homeostasis. To maintain this important function, vascular endothelial cells replenish themselves to remove damaged cells, when damage occurs. However, endothelial cell senescence significantly reduces the capacity of the vascular endothelium to self-repair, leading to overall decline in endothelial cell number and/or function. Accumulation of senescent endothelial cells in the vascular bed results in increased expression of senescence-associated secretory phenotypes (SASPs), which also results in the senescence of adjacent cells in the vasculature. Eventually, vascular rarefaction and dysfunction occurs as risk factors and/or progression of many human diseases [162].

Atherosclerosis is a complex, chronic inflammatory disease affecting large- and medium-sized arteries. Chief among the factors responsible for the pathogenesis of atherosclerosis (including inflammation, hyperlipidemia, and lipid peroxidation) is cellular senescence. Factors such as hyperglycemia and oxidative stress cause premature cell senescence due to reduction in telomerase activity and consequently telomere shortening, mitochondrial dysfunction, exacerbated ROS production, and DNA damage. Atherosclosis due to cellular senescence is a leading cause of cardiovascular disease-related morbidity and mortality. In the human vasculature, senescent endothelial cells and smooth muscle cells and macrophages are critically involved in atherosclerosis development [163,164]. Senescence-associated endothelial dysfunction is known to be the first stage of atherosclerosis development due to changes in the functional phenotype and/or secretome of senescent endothelial cells. Typically, this shift results in reduced production of vasodilatory factors such as nitric oxide, prostacyclin, hydrogen sulfide, and increased production of inflammatory and vasoconstrictive factors such as reactive oxygen species, endothelin-1, and angiotensin II. In addition, senescent endothelial cells have altered metabolic activities such as glycolysis, oxidative phosphorylation, and fatty acid oxidation. Altogether, these shifts in the functional and metabolic phenotypes of aged endothelial cells contribute significantly to vascular aging and/or atherosclerosis [93,165].

A shift in functional and secretory phenotypes of senescent endothelial cells favors excessive vasoconstriction and hypertension. Oxidative stress, basically referring to an imbalance between nitric oxide production and reactive oxygen species generation, is a key contributor to endothelial dysfunction. An increase in ROS inhibits eNOS mRNA expression and eNOS activity via activation of the PI3K/Ras/Akt/MAPK pathway. Reduction in nitric oxide bioavailability due to aged and dysfunctional endothelial cells, as well as inhibition by ROS, is a major contributor to endothelial dysfunction in aging models of both murine models and human subjects. Thus, the eventual outcome of senescence-associated endothelial dysfunction during aging is typically hypertension. Paradoxically, aging, as well as aging-associated hypertension and hypertension per se, either independently or collectively, impairs endothelial function, leading to atherosclerosis, resulting in cardiovascular and cerebrovascular outcomes [166].

The blood−brain barrier (BBB) is the physical barrier and regulator of transport of molecules between the brain and the rest of the body. Disruption of the BBB and uncoupling of the neuro-vasculature, especially in aged subjects, are hallmarks of cerebrovascular diseases and cognitive decline. Cerebral endothelial cells form the core of the BBB. Thus, senescence-associated dysfunctional cerebral endothelial cells are a direct cause of neuro-vascular disease and cognitive decline among the elderly. Changes to endothelial nitric oxide synthase (eNOS)/nitric oxide (NO) signaling, tight junctions, angiogenesis, and neuroinflammation in cerebral endothelial cells have the potential to drive cerebrovascular dysfunction. NO bioavailability is a crucial regulator of normal cerebral endothelial cell function and regulates cerebral blood flow in response to changes in cellular energy demands. Decreased NO bioavailability via decreased synthesis or accumulation of ROS leads to ineffective cerebral blood flow regulation and hypoperfusion of the brain and ultimately contributes to neuronal cell death and cognitive dysfunction. Decreased angiogenesis occurs with aging and in cerebrovascular disease, which can lead to hypoperfusion, impaired adaptation to hypoxia, compromised recovery to tissue damage, and exacerbated ischemic tissue injury [167].

## 6. Senolytic Therapy Targeting Endothelial Senescence

Accumulation of senescent endothelial cells leads to diseases associated with aging. Available evidence underpins a central role of endothelial senescence in vascular aging and suggests that targeting various senescent pathways in endothelial cells is a viable entry point to promote healthy aging. Different therapeutic strategies can be used to prevent endothelial senescence or eliminate senescent endothelial cells, thus enabling rejuvenation. Senescent endothelial cells exhibit specific functional abnormalities, such as loss of proteostasis and genomic stability, reduced nitric oxide (NO) production, increased expression of pro-inflammatory cytokines and oxidative stress, abnormal autophagic flux, mitochondrial dysfunction, and deregulation of sirtuins [7,168,169,170,171]. Experimental interventions specifically targeting senescent endothelial cells and vascular aging have shown promising results in model organisms. Telomere uncapping or telomere shortening, for example, induces an early-onset cardiovascular aging phenotype in mice, characterized by premature myocardial hypertrophy, fibrosis, and diastolic dysfunction, as well as vascular oxidative stress, endothelial dysfunction, and increased blood pressure [172,173]. Overexpression of human telomerase reverse transcriptase (hTERT) in HGPS endothelial cells extends telomere length, restores endothelial function and gene expression, increases sirtuin 1 (SIRT1) expression, reduces markers of DNA damage and extends life span [174,175]. Similarly, targeting the vascular endothelial growth factor (VEGF) signaling prevents age-related microvascular attrition and effectively delays age-related pathologies across various organ systems, resulting in a prolonged lifespan of mice [176]. Endothelial overexpression of the longevity regulator SIRT1 attenuates age-related decline in endothelium-dependent vasodilatation, hypertension, atherosclerosis, and various organ dysfunctions [177,178,179,180].

Vascular endothelial cells, particularly those in the cerebral microcirculation, are exposed to senescence-inducing stimuli such as biochemical and hemodynamic factors, as well as numerous senolytic drugs. Long-term treatment with senolytic drugs, dasatinib (a Src/tyrosine kinase inhibitor)/quercetin (a natural flavonoid that binds to BCL-2 and BCL-XL) delays vascular aging as indicated by improved endothelium-dependent vasodilatation, as well as reduced aortic calcification and osteogenic signaling in hypercholesterolemic mice [181]. Dasatinib eliminates senescent human fat cell progenitors, while quercetin is more effective against senescent human endothelial cells. The combination treatment can slow endothelial cell senescence in humans [182]. Senescent endothelial cells are particularly susceptible to apoptosis induced by ABT263/Navitoclax, an experimental orally active anti-cancer drug which inhibits the apoptosis regulator proteins Bcl-2 and Bcl-XL [183]. Treatment with ABT263/Navitoclax enhances neurovascular coupling and improves hippocampus-encoded functions of learning and memory in aged mice [184]. Similar improvements in cognitive function have been observed in elderly INK-ATTAC mice treated with AP20187, a drug that eliminates p16^Ink4a^-positive senescent cells, or the senolytic treatment combination dasatinib/quercetin [185]. Senescent cell burden can be safely reduced in a clinical context [186]. Aerobic exercise prevents endothelial senescence and protects against endothelial dysfunction in older adults [187]. In fact, the effects of cardiac glycosides are partially attributable to their senolytic activity [188,189,190]. On the other hand, rapamycin, a macrolide antibiotic, inhibits the mammalian target of rapamycin (mTOR), a protein kinase which nucleates two distinct signaling complexes, known as mTORC1 and mTORC2 [191]. Rapamycin suppresses the mammalian TORC1 complex, which regulates translation, and extends lifespan in diverse species, including mice [192]. Treatment of rapamycin has persistent effects that can robustly delay aging, influence cancer prevalence and enhance longevity and healthspan [193,194]. Dietary intake of rapamycin was shown to reverse age-related vascular dysfunction and oxidative stress, in association with reduced arterial expression of the senescence marker p19 [92]. Rapamycin was also found to decrease IL6 and other cytokine mRNA levels while selectively suppressing translation of the membrane-bound cytokine IL1A. Reduced IL1A diminishes NF-κB transcriptional activity, which regulates much of the SASP; exogenous IL1A restores IL6 secretion to rapamycin-treated cells [195].

Considering that senescent cells take up to seven days or longer to accumulate and develop SASP, most therapeutics are adopting a “hit-and-run” strategy that does not require daily or weekly administration. The first generation of senolytic drugs was developed, aiming to disrupt the senescent cell anti-apoptotic pathways (SCAPs) and other pro-survival signaling network [182,196,197,198,199]. Currently, senolytics have expanded and include a broad range of senescent features as targets. However, these senolytic strategies may elicit off-target effects or interfere with beneficial populations [200,201,202]. There is an urgent demand for innovative senolytic drugs devoid of side effects for clinical applications. In general, the phenotype of endothelial senescence depends on the context of stress and the surrounding microenvironment, thus varied by cell location and tissue of origin and/or residence. As a result, the functional impact of senolytic therapy is influenced by how and where senescence is induced. For example, treatment with the BCL-2/BCL-xL inhibitor senolytic drug ABT263/Navitoclax improves functional hyperemia in the brain of aged mice but has no effects on endothelium-dependent acetylcholine-induced relaxation of aorta rings [184]. The differential effects of senolytic treatment may be attributable to the presence of more senescent endothelial cells in the aged cerebral microvasculature than in the conduit arteries [203]. Thus, organ specificity needs to be considered for the treatment of diseases, such as HFpEF. In summary, since senescent cells are highly heterogeneous, targeted strategies are needed that ideally preserve senescent cells in beneficial contexts while eliminating effects that are detrimental [204].

## 7. Challenges and Perspectives

Senescent endothelial cells promote the development of age-related disorders, including cardiometabolic and neurodegenerative diseases, whereas suppression of endothelial senescence ameliorates phenotypic features of aging in various models. However, the occurrence, location, function, and impact of senescent endothelial cells are very heterogeneous and depend on the tissue environment as well as the type and duration of stress/injury [205]. SASP also varies, depending on the type of endothelial cells, surrounding tissue, and triggers of senescence. Moreover, transient presence of senescent cells is beneficial for conditions such as normal development, wound healing, tissue regeneration, and cancer prevention [206]. Thus, targeted approaches are needed to preserve senescent cells in beneficial contexts while eliminating those that are detrimental. Specific depletion of senescent endothelial cells in a context-dependent manner may be a more promising strategy for the treatment of age-related diseases [207]. To this end, understanding the heterogeneity in both the molecular biology and the pathophysiological function of senescent endothelial cells facilitates the identification of specific biomarkers for targeted senolytic therapy [19,94,95,204,208]. Another issue remaining to be explored is the potential side effects of senolytic treatment [209]. Without a transcriptional analysis at the single-cell level, it is difficult to distinguish between different types of senescent endothelial cells. To identify the most suitable senolytic agents, optimize the dosage for administration and determine the combinations for treatment of various conditions, new mouse models to induce genetic elimination of specific senescent cell types or test cell type-specific senolytics should be generated. Potential gender differences are another critical consideration. The goal is to explore and develop senolytic agents that act on specific senescent mechanism(s) and types of endothelial cells or tissues for selective depletion of pathological senescent cells and reduce the “bystander effect” and “off-target toxicities”.

## Figures and Tables

**Figure 1 ijms-25-01978-f001:**
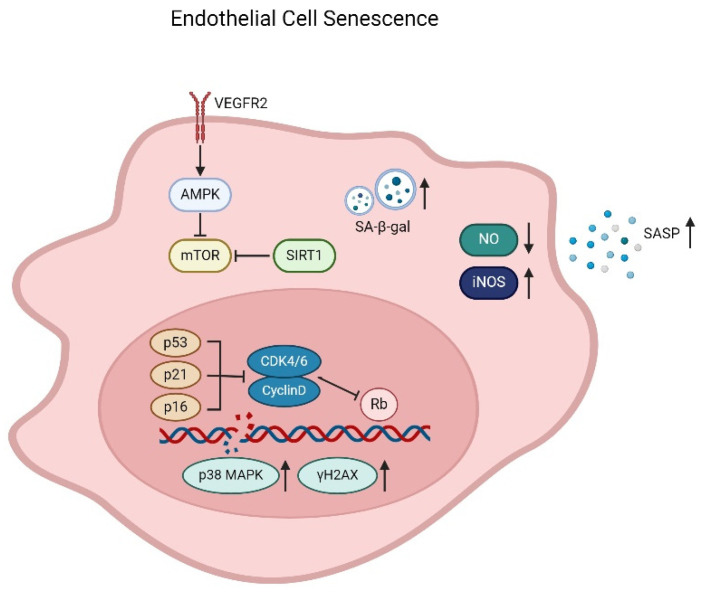
Mechanism of endothelial cell senescence. Senescent endothelial cells develop a senescence-associated secretory phenotype (SASP) and an increase of SA-β-Gal activity. Senescence results in decrease of endothelial NO synthase expression and reduction of NO production, and the inducible NO synthase (iNOS) and ROS levels increase significantly, leading to DNA damage and promotion of senescence and apoptosis. Senescent cells also exhibit upregulation of p53/p21 or p16/Rb pathways and markers associated with activation of the DNA damage response (p38MAPK and γH2AX). In addition, the VEGF signaling pathway, along with AMPK, mTOR, and SIRT1, is also involved in the endothelial cell senescence.

**Figure 2 ijms-25-01978-f002:**
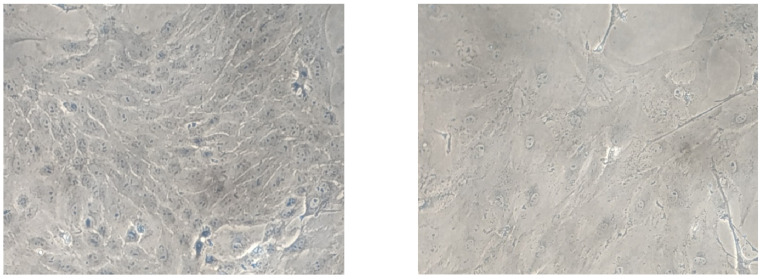
Comparison of young and senescent endothelial cells isolated from porcine aorta (Magnification: 4×). Arteries were separated from the heart and immediately placed in freshly prepared cold Earle’s balanced salt solution (EBSS). After clearing all the peripheral connective tissues, aortas were cut longitudinally and pinned down for en face cell isolation in a glass dissecting dish. A sufficient cold EBSS was added to rinse the aortic fragments by gentle shaking. The EBSS was discarded and replaced rapidly by a fresh EBSS to reduce the exposure time to air in order to enhance cell viability. After thorough washing for five times, the EBSS was removed completely before collecting the endothelial cells. Endothelial cells were collected gently from the luminal surface of aorta by scraping with a scalpel blade and collected in pre-warmed DMEM containing 15% fetal bovine serum (FBS) and 1% penicillin-streptomycin-fungizone (PSF). The cell pellets were washed by centrifugation (200× *g*, 5 min) and re-suspension for two times. Finally, the PAECs were resuspended by the pre-warmed DMEM containing 15% FBS and 1% PSF and transferred into a 60 mm 1% gelatin-coated Petri dish and cultured in DMEM with 15% FBS and 1% PSF in a humidified incubator containing 5% CO_2_/95% O_2_ at 37 °C. The medium was replaced every two days. The young cells (**left**) show a typical “cobblestone” monolayer, whereas after 4-week culture, the senescent (**right**) endothelial cells exhibit an enlarged, multinucleated and flattened morphology [102].

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
