# Peer review of "Endothelial Senescence: From Macro- to Micro-Vasculature and Its Implications on Cardiovascular Health"

_ijms, 2024, doi:10.3390/ijms25041978_

Round 1
Reviewer 1 Report
Comments and Suggestions for Authors
Thank you for allowing me to review this impressive piece of work.
General:
The paper reads (very) well and is most informative regarding the (patho)physiology. There are some minor typo's throughout the text, please check.
Abbreviations:
A much appreciated listing. Throughout the text full names are systematically followed by the abbreviation; this is done well. Suggest to include ROS (reactive oxygen species) in the listing as well. Please check for other (returning) abbreviations throughout the text that may be included in the listing.
Page 2. The definition of senescence is explained excellently, but in a very complex first sentence that may not benefit further reading. Why not, as a suggestion, define it as cessation of cell division. Than follow with the existing sentences. The first 2 sentences than read: 'Cellular senescence is the cessation of cell division. Senescence is originally defined as the irreversible loss of proliferative potential in somatic cells, which enter a viable and metabolically active state of permanent growth arrest that is distinct from quiescence and terminal differentiation [1-3]. etc., etc.'
The title could be extended, for example 'Endothelial Senescence: from macro- to micro-vasculature and its consequences on cardiovascular health' (or 'disease'). This may attract a broader (clinically oriented) audience.
I have no further suggestions on this well written manuscript.
Comments on the Quality of English Language
See the comments section. Just minor issues.
Author Response
Thank you for your comments and suggestions. We have revised the manuscript accordingly and all changes are marked as red.
Abbreviations: ROS (reactive oxygen species) was included in the list. Other abbreviations were also added to the list.
Page 2. The first two sentences of introduction were revised according to your suggestion.
The title was revised as “Endothelial Senescence: from macro- to micro-vasculature and its implications on cardiovascular health”.
Reviewer 2 Report
Comments and Suggestions for Authors
The present review focuses on the process of senescence in endothelial cells. Key pathways being involved are described, the changes of endothelial cells during aging-associated disorders such as obesity, diabetes mellitus, COVID-19, heart failure are discussed, as well as senolytic therapeutic strategies are presented.
The review is well structured, and is based on the extensive number of references. However, the number of the state-of-the art articles, dated within the recent 5 years, may be extended.
Other suggestions to improve the manuscript include the followings:
1. The role of endothelial progenitor cells (EPCs) in endothelial senescence should be mentioned.
2. Line 63: extracellular vesicles per se are secreted both by the young and senescent cells, but biological properties of EVs from aging cells become compromised and acquire senescence-associated phenotype. Please, specify it here.
3. Line 279 – KDR is not deciphered, as well as Mfsd2a protein (Line 288)
1. Line 427 – please, specify that the function of ERCC1 is endonuclease
Comments on the Quality of English LanguageEnglish is fine, except a few typos/mistakes:
Line 84: role IN peripheral vascular resistance..
Line 107: OF aged microvasculature
Line 118: has openEd
423 - atrial
124: IS abound
176: DUE
392: changes
Author Response
Thank you for your comments and suggestions. We have revised the manuscript accordingly and all changes are marked as red.
- The role of endothelial progenitor cells (EPCs) in endothelial senescence should be mentioned.
The role of EPCs in endothelial regeneration and maintenance in ageing was described in page 5.
- Line 63: extracellular vesicles per se are secreted both by the young and senescent cells, but biological properties of EVs from aging cells become compromised and acquire senescence-associated phenotype. Please, specify it here.
Corresponding sentences were revised:” By secreting a plethora of factors, including proinflammatory cytokines, chemokines, growth modulators, matrix metalloproteinases and compromised extracellular vesicles, such as exosomes which represent senescence-associated phenotype, senescent cells reprogram the surrounding microenvironment and cause tissue damage, thus promoting ageing and the development of age-associated diseases.”
- Line 279 – KDR is not deciphered, as well as Mfsd2a protein (Line 288)
Full names of these abbreviations were added.
- Line 427 – please, specify that the function of ERCC1 is endonuclease
Corresponding sentences were revised:” Excision repair cross-complementation group 1 (ERCC1), an endonuclease non-catalytic subunit, is an essential component in the DNA damage repairing system [100].”
A few typos were revised accordingly.